# Fused-Deposition Modeling 3D Printing of Short-Cut Carbon-Fiber-Reinforced PA6 Composites for Strengthening, Toughening, and Light Weighting

**DOI:** 10.3390/polym15183722

**Published:** 2023-09-11

**Authors:** Bin Sun, Suhail Mubarak, Guocun Zhang, Kangming Peng, Xueling Hu, Qia Zhang, Lixin Wu, Jianlei Wang

**Affiliations:** 1CAS Key Laboratory of Design and Assembly of Functional Nanostructures, Fujian Key Laboratory of Nanomaterials, Fujian Institute of Research on the Structure of Matter, Chinese Academy of Sciences, Fuzhou 350002, China; loney_bin@163.com (B.S.);; 2University of Chinese Academy of Sciences, Beijing 100049, China; 3CAS Haixi Industrial Technology Innovation Center in Beilun, Ningbo 315830, China; 4State Key Laboratory of Fluid Power & Mechatronic System, Key Laboratory of Soft Machines and Smart Devices of Zhejiang Province, Center for X-Mechanics, Department of Engineering Mechanics, Zhejiang University, Hangzhou 310027, China; 5School of Automotive Engineering, Dalian University of Technology, Dalian 116024, China; 6College of Chemistry and Materials Science, Fujian Normal University, Fuzhou 350007, China; 7College of Chemistry, Fuzhou University, Fuzhou 350116, China; 8Chunhui Technology Group Co., Ltd., Fuzhou 350019, China

**Keywords:** polyamide, fiber reinforced composites, fused deposition 3D printing, heat treatment process, orientation distribution, crash box, honeycomb structure

## Abstract

Additive manufacturing of carbon-fiber-reinforced polymer (CFRP) has been widely used in many fields. However, issues such as inconsistent fiber orientation distribution and void formation during the layer stacking process have hindered the further optimization of the composite material’s performance. This study aimed to address these challenges by conducting a comprehensive investigation into the influence of carbon fiber content and printing parameters on the micro-morphology, thermal properties, and mechanical properties of PA6-CF composites. Additionally, a heat treatment process was proposed to enhance the interlayer bonding and tensile properties of the printed composites in the printing direction. The experimental results demonstrate that the PA6-CF25 composite achieved the highest tensile strength of 163 MPa under optimal heat treatment conditions: 120 °C for 7.5 h. This corresponds to a significant tensile strength enhancement of 406% compared to the unreinforced composites, which represents the highest reported improvement in the current field of CFRP-fused deposition 3D printing. Additionally, we have innovatively developed a single-layer monofilament CF-OD model to quantitatively analyze the influence of fiber orientation distribution on the properties of the composite material. Under specific heat treatment conditions, the sample exhibits an average orientation angle μ of 0.43 and an orientation angle variance of 8.02. The peak frequency of fiber orientation closely aligns with 0°, which corresponds to the printing direction. Finally, the study explored the lightweight applications of the composite material, showcasing the impressive specific energy absorption (*SEA*) value of 17,800 J/kg when implementing 3D-printed PA6-CF composites as fillers in automobile crash boxes.

## 1. Introduction

Fiber-reinforced composites (FRP) are derived from the modification of polymers with either continuous or short fibers [1,2,3,4,5,6]. By combining stiff and strong fibers with lightweight polymer matrices, fiber-reinforced composites offer a high strength-to-weight ratio, excellent processability, and resistance to wear [7,8,9]. Polyamide 6 (PA6), as a semi-crystalline polymer, possesses excellent processability and shape ability, along with outstanding wear and corrosion resistance, rendering it an ideal choice for a polymer matrix [10]. For instance, Beom et al. [11] enhanced the carbon fiber/PA6 composite via interfacial reinforcement using a mixture of adhesive and reduced-graphene oxide coating. The modified composite mixture exhibited a 73% increase in flexural strength and an 84% increase in modulus. Federico et al. [12] processed carbon fiber-reinforced PA6 composites using Fused-Deposition Modeling (FDM) and varied the raster angle in the layer sequence according to different printing architectures. Laser writing was then performed to obtain conductive tracks on the surface of the printed parts. Furthermore, reducing fiber content or incorporating plasticizers enhanced the toughness of the composite material. Elena et al. [13] demonstrated that increasing the fiber content in carbon fiber-reinforced PA6 composites (10–30%) resulted in an increase in internal porosity and improved mechanical properties.

Composites can be reshaped through molding or extrusion processes, which offer advantages in mass production [14]. Material extrusion-based fused-deposition modeling (FDM) has emerged as a popular additive manufacturing technology, owing to its extensive design flexibility and a wide range of material options [12,15,16,17,18,19,20,21,22]. Consequently, volume contraction occurs, resulting in diminished interlayer adhesion and an increased porosity in the printed parts [23,24]. Researchers have sought to address the inherent limitations of fused-deposition 3D printing by optimizing printing parameters [25,26]. However, these efforts have not yielded significant improvements in the mechanical performance of printed materials [27,28]. In this study, we effectively addressed the inherent issues of FDM by adjusting fiber content, optimizing printing parameters, and optimizing the heat treatment process, significantly improving the mechanical performance of the composite materials.

On the other hand, composite materials manufactured through fused-deposition 3D printing often lack precise control over the fiber orientation of the parts [7,29,30]. It is crucial to control the distribution of fiber orientation to enhance the performance of composites [31]. Existing fiber alignment techniques primarily involve adjusting printing parameters, applying external forces, and developing predictive models [32,33,34,35].For instance, Yan et al. [36] quantified the impact of printing parameters on the fiber orientation distribution of carbon-fiber-reinforced ABS/PLA composites, demonstrating the significant influence of filament line width and nozzle height on the orientation distribution of the composite fibers. Pasita et al. [37] developed an anisotropic viscous flow model to predict fiber orientation, microbead geometry, and inter-bead contact interface geometry. However, there is limited research on characterizing fiber orientation within a single filament. To address this research gap, we designed a simplified model of a single-layer monofilament CF-OD structure to mitigate the effects of other parameters and external conditions on the fiber orientation of composites. To accurately assess the fiber distribution in 3D-printed PA6-CF composites, we performed a meticulous process involving the careful removal of the upper and lower surfaces of the obtained monofilament samples. Subsequently, milling and polishing techniques were employed on the exposed specimens to determine the orientation distribution of the internal structural fibers.

In summary, the layer-by-layer nature of FDM printing inevitably leads to weak interlayer bonding in printed composites. Moreover, the incorporation of fibers increases the material’s porosity, while the distribution of fiber orientation limits further improvement in material properties. This study explores the potential of carbon-fiber-reinforced PA6 composites under FDM printing conditions. We prepared PA6-CF-printed samples by adjusting the fiber content, optimizing printing parameters, and optimizing the heat treatment process. The fiber orientation distribution was quantified using a single-layer monofilament CF-OD model, and the mechanism underlying the improvement in mechanical properties with variations in microstructure and thermal properties was analyzed. Additionally, this study explored the lightweight applications of the composite material.

## 2. Materials and Methods

### 2.1. Materials

Polyamide 6 (PA6 J-2400, density 1.151 g/cm^3^), manufactured by Polymer Shun New Material Co. Ltd., (Hangzhou, Zhejiang, China) was utilized as the polymer matrix material. Short-cut carbon fibers were procured from Tokuyama Carbon Fiber Co. (Tokuyama, Japan). The characteristics of the carbon fibers are as follows: a fiber diameter of 7 μm, a fiber density of 1.502 g/cm^3^, a tensile strength of 3550 MPa, and a tensile modulus of 230 GPa. Silane coupling agent (KH-550, Wanhua Chemical Company, Yantai, China) and 95% ethanol were used without further purification.

### 2.2. Preparation of PA6-CF Composite Filaments

PA6 raw pellets were soaked overnight with short carbon fibers in a mixture of 95% alcohol and a silane coupling agent to enhance the interfacial adhesion between the polymer matrix and the carbon fibers [10]. The resulting silanized masterbatch was co-blended using the HAAKE twin-screw extruder to extrude filaments, which were then collected and pelletized by the downstream winder. This process was repeated twice to achieve a uniform dispersion of carbon fibers in the PA6 matrix, as shown in Figure 1Ab. Three PA6-CF composites with varying carbon fiber contents were prepared by adjusting the rotational speed of the extruder screw and the PA6 feed rate [38]. The carbon fiber mass fractions of these composites were determined through thermogravimetric analysis experiments (30–900 °C, 10 °C/min, N2). In this study, the composites were designated as PA6-CF10, PA6-CF20, and PA6-CF25. The PA6-CF filaments were extruded using a single-screw extruder after further blending, with the filament diameter controlled at 1.75 ± 0.05 mm. During the extrusion process, the barrel and die head temperatures were set at 240 °C and 250 °C, respectively, while the screw speed remained constant at 50 rpm [13,39].

### 2.3. 3D Printing and Heat Treatment Process of PA6-CF Composites

Composite filaments need to be dried in an oven at 75 °C for 12 h to remove moisture before printing or characterization. The custom-built 3D printer (E2CF, Raised 3D) is specifically designed for FDM printing of fiber-reinforced composites because of several distinct advantages: (1) Dual gear feeding system. Smoother printing for the lack of fluidity of carbon fiber composites. (2) Highly durable nozzle. The nozzle hardness is increased from hrc30 to 60 to withstand the wear and tear of carbon fiber on the nozzle. In this work, the temperature and time of heat treatment are considered as two basic variables. The heat treatment process is conducted in a vacuum drying oven, and the vacuum pressure is maintained at 10 MPa throughout the entire experimental process. The heat-treated samples are not allowed to contact with metal or each other. The variation interval of the heat treatment temperature is determined by the glass transition temperature Tg of the printed composite PA6-CF, while the variation interval of the heat treatment time is determined by the pre-experiment. The trends of changes in mechanical properties, micro-morphology, fiber orientation division and crystallinity of the printed composites were analyzed under 5 h of heat treatment. Finally, the heat treatment temperature parameters were optimized to be room temperature, 90 °C, 120 °C, and 150 °C, and the heat treatment time was optimized to be 0 h, 5 h, 7.5 h, and 10 h. The 100 mm× 30 mm× 5 mm sample printed from PA6-CF composite material is used as the standard specimen. The dimensional changes of the standard specimen after heat treatment are measured precisely using a vernier caliper, with a measurement precision of 0.01 mm. The printing parameters of the E2CF printer and heat treatment parameters are shown in Table 1.

### 2.4. Single-Layer Monofilament CF-OD Model and Fiber Orientation Testing Methods

We utilized an optical microscope (Leica DM750) at 20× magnification to observe the distribution of planar fibers. The orientation angle θ and the number of individual fibers were recorded using ImageJ and subjected to statistical analysis.

### 2.5. Characterization of Material Micromorphology

The dispersion of carbon fibers (CF) in the composites and the microstructure of the fractured tensile members were characterized using a scanning electron microscope (SU3, Hitachi, Tokyo, Japan, 15 kV). The fiber orientation distribution in the CF-OD model was also characterized using an optical microscope (T2-HD206, AOSVI, Suzhou, China). For specimens with fiber orientation distribution on the surface, acetone treatment was applied to expose the carbon fibers. Meanwhile, specimens with fiber orientation inside the CF-OD were ground and polished. Grinding was performed using 1000-mesh sandpaper, followed by polishing with a 3 μm polycrystalline diamond suspension (S01-1104D1, Ningbo, China). To determine the length distribution of the fibers in the composite prints, the composites were burned in a muffle furnace, and the residual CF was observed under a microscope. The density and porosity of the composites were calculated using Archimedes’ law and ASTM D3171-15 standard test method.

### 2.6. Characterization of Thermal Properties of Materials

The stability of the composites was assessed using the STA4493F3 thermogravimetric analyzer, with the mass fraction of CF calibrated. The temperature was incrementally increased at a rate of 10 °C/min within the test temperature range of rt-900 °C. In addition, the composites’ crystallization behavior during the heat treatment process and the variations in their melting temperature (Tm) and glass transition temperature (Tg) were analyzed using a differential scanning calorimeter (TA Instruments Q2000). Under a nitrogen atmosphere, approximately 10 mg of dried samples were cycled between 30 °C–360 °C–90 °C, with a temperature ramp rate of 10 °C/min for both heating and cooling stages.

### 2.7. Characterization of Mechanical Properties of Materials

The tensile and flexural strengths of the composites were evaluated using a universal testing machine (CMT6104, China) with loading rates of 5 mm/min and 2 mm/min, respectively. The interlaminar shear strength (ILLS) was determined with reference to the literature test methods [28]. The impact strength of the materials was determined using a cantilever beam impact test performed with a Charpy pendulum impact tester (HST-800, China) equipped with a 50.7 J pendulum.

## 3. Results

### 3.1. Effect of CF Content and Printing Parameters on the Properties of PA6-CF Composites

#### 3.1.1. CF Content

To explore the potential of fused-deposition 3D printing for short-cut carbon-fiber-reinforced composites, we conducted a study on the influence of carbon fiber mass fraction on the properties of the composites. The thermal stability of PA6-CF was characterized using thermogravimetric analysis (TGA), revealing an initial degradation temperature (Td) of 389.1 °C for pure PA6 material, as illustrated in Figure 2a. The mass fraction of short-cut carbon fibers exhibited a positive correlation with the Td temperature of the composites, increasing to 391.2 °C, 399.3 °C, and 405 °C, respectively, accompanied by varying degrees of elevation in the temperature corresponding to the maximum weight loss rate. The observed phenomenon can be ascribed to the significantly higher decomposition temperature of the fibers in comparison to the matrix [40]. Furthermore, the robust interfacial interaction between the fiber and the matrix impedes the mobility and degradation of the polymer chains during the thermal decomposition phase [41]. The fiber content in the three composites was determined by calculating the difference in weight percentage of residue at 900 °C compared to pure PA6. These composites were approximately denoted as PA6-10, PA6-20, and PA6-25 in this study. Figure 2b,c illustrates the DSC melting and crystallization curves of PA6-CF composites with varying fiber mass fractions. These curves provide valuable thermal parameters, including the melting temperature (Tm), crystallization temperature (Tc), and percent crystallization (*X_c_*). *X_c_* can be calculated using the following equation:(1)$$X_{c}=\frac{∆H_{m}}{\left(1-w\right)∆H_{m,PA6}^{0}}\times 100\%$$
where Δ*H_m_* is the enthalpy of melting, *w* denotes the weight fraction of fibers, and $∆H_{m,PA6}^{0}$ denotes the theoretical enthalpy of melting for 100% pure crystallization.

The addition of carbon fibers enhances the mechanical strength and stiffness of the composite [42]. At elevated temperatures, this reinforcement effect effectively withstands the melting of PA6 polymer molecules, requiring higher temperatures for achieving a molten state. The melting temperature Tm of PA6-CF10 is 245 °C, whereas PA6-CF20 and PA6-25 exhibit a respective increase in melting temperature by 2 °C and 5 °C. Furthermore, the surfaces of carbon fibers possess high surface energy and reactive groups, thus augmenting the heterogeneous nucleation capacity of the composites. These fibers can serve as crystallization nuclei, promoting the crystallization of polymer molecules. Based on the calculated *X_c_* (crystallinity) results, the crystallinity of the composites initially increases and then decreases with the escalating content of carbon fibers: *X_c_* values of 27.89%, 31.21%, and 30.36% were obtained for the respective composites. This phenomenon can be ascribed to the robust interfacial interactions between the silanized fibers and the matrix when the mass fraction of carbon fibers reaches 25%. These interactions hinder the mobility of polymer chains. As a semi-crystalline polymer, a portion of the PA6 polymer chain segments align into ordered crystalline regions through the crystallization process, while the presence of carbon fibers impacts the distribution of the remaining disordered polymer chain segments.

Figure 3f illustrates the stress–strain curves depicting the tensile properties of CF composite prints with varying contents. The tensile strength of pure PA6 is 33 MPa, whereas PA6-CF10 prints exhibit an elevated tensile strength of 58 MPa. PA6-CF25 prints achieve a tensile strength of 93 MPa, resulting in a 1.82-fold increase in strength, albeit with a reduction in elongation at break to 9.2%. To elucidate the pattern of tensile strength variation in the composite prints, scanning electron microscopy (SEM) was employed to observe the fiber dispersion in the composites, as shown in Figure 3a–c. At low CF content, carbon fibers were disorderedly arranged. However, higher CF content leads to a more ordered arrangement of carbon fibers, enhancing the rigidity and strength of the composites. Moreover, with an increase in carbon fiber content, the interfacial region between the carbon fibers and the polymer matrix expands. This enlarged interfacial bonding region facilitates improved stress transfer and the ability to impede crack propagation, consequently enhancing the mechanical properties of the composite [32]. SEM images of the tensile fracture provide a clearer visualization of this phenomenon, as shown in Figure 4. In the composite with 10 wt% CF content, only small voids and holes pulled out by carbon fibers are observed. However, in composites with 15 wt% CF content, the number of carbon fiber distributions and the amount of carbon fibers pulled out along the tensile direction significantly increase. When the CF content reaches 20 wt%, the entire cross-section exhibits a denser structure with tighter interfacial bonding, although some unimpregnated carbon fibers are also present. 

The composites were burned in a muffle furnace to obtain the length distribution of the fibers in the composite prints. The residual carbon fibers (CF) were observed under a microscope, as depicted in Figure 3d. Following the burning process, the majority of carbon fibers in the matrix retained their original morphology, with only a few transitioning into powder form. The carbon fibers that preserved their morphology were quantitatively evaluated by recording their length and quantity. The length distribution of approximately 1200 residual CFs is illustrated in Figure 3e, wherein the majority of fibers fell within the range of 200–500 μm, aligning with microscopy and scanning electron microscopy (SEM) findings. The average value of the fiber length distribution was 345 μm, with fibers ranging from 340 μm to 350 μm accounting for 19% of the total count. Moreover, the cumulative frequency curve of the fiber length distribution exhibited an S-shape, indicating that carbon fibers with this length distribution contribute to enhanced mechanical strength in the material.

#### 3.1.2. Printing Parameters

In addition, to further enhance the mechanical properties of the composites, we investigated the impact of printing parameters, such as grating angle and substrate temperature, on the properties of PA6-CF-printed parts. During the printing process, variations in the grating angle influence the relationship between fiber orientation and tensile direction. By enlarging the images of the stretched sample surface under a microscope (refer to Figure 5a–c), it is evident that distinct grating angles result in different deposition directions of adjacent beads. However, there is a close arrangement between adjacent beads and individual fibers, with no presence of voids. As presented in Appendix A and Table 2, the outcomes for PA6-CF samples at different grating angles align with the stretching direction. Notably, samples printed with a grating angle of 0° demonstrated the highest tensile strength and elongation at break, followed by those at 60°, 45°, 135°, and 90°. For instance, when all other conditions were held constant, the sample with a grating angle of 0° exhibited a tensile strength of 97.7 MPa and a tensile modulus of 1021.8 MPa, while the sample with a grating angle of 90° showed a tensile strength of 77.7 MPa and a tensile modulus of 711.5 MPa. Moreover, elevating the base plate temperature resulted in an improvement in the mechanical properties of the samples. Taking the PA6-CF25 composite as an example, for samples with all grating angles set to 0°, the flexural strength was 111.2 MPa and the impact strength was 17.3 kJ/m^2^ at a base plate temperature of 60 °C. However, at a base plate temperature of 90 °C, the flexural and impact strengths underwent an increase of 11% and 45%, respectively, reaching 123.1 MPa and 25.1 kJ/m^2^.

Table 2 presents the values of mechanical properties for the 3D-printed composite PA6-CF, considering three variables: CF content, raster angle, and print substrate temperature. The table encompasses the results of 16 cross combinations and six categories of mechanical properties tested. Notably, a raster angle of 90° leads to a significant increase in the flexural strength of the 3D-printed material, with the highest value reaching 145.2 MPa. Furthermore, impact strength shows a positive correlation with CF content and print substrate temperature, where the results from testing the PA6-CF25 composite at a print substrate temperature of 90 °C and a raster angle of 90° demonstrate the highest impact strength of 26.1 kJ/m^2^. In order to provide a clearer explanation of the influence of the three factors on the mechanical performance of the composite material, we have included the results of the three-factor orthogonal experiment in Appendix A. The results indicate that fiber content is the key factor affecting the tensile strength and impact strength of the composite material, while print substrate temperature is the key factor affecting the flexural strength of the composite material. The mechanical properties of PA6-CF composites were effectively improved by altering the carbon fiber content and adjusting the printing parameters. However, electron micrographs revealed the presence of inconsistent fiber distribution and interlayer voids within the material, which acted as obstacles to further enhancing the composite properties.

### 3.2. Analysis of the Effect of Heat Treatment Process on Mechanical Properties

#### 3.2.1. Effect of Heat Treatment Process on Fiber Orientation of 3D-Printed Composites

3D-printed fiber-reinforced composites often exhibit microstructural anisotropy and mechanical property anisotropy [1]. To mitigate the negative effects of anisotropy, we developed a CF-OD model with a single layer of monofilaments for testing fiber orientation. The CF-OD model was created by printing four monofilaments along a square wave path, with each monofilament having a line width of 0.4 mm, a height of 0.15 mm, and a length of 50 mm, ensuring no gaps theoretically.

Figure 6 illustrates the microscopic images of single-fiber CF-OD models printed under various heat treatment conditions. To aid observation and analysis, white dashed lines highlight the carbon fibers. In Figure 7a, the printed composite with a raster angle of 0° exhibited well-defined boundaries for individual fibers after undergoing a 5-h treatment at 120 °C. The black areas in the image represent voids. Although the quantity of carbon fibers at the fiber edges was relatively lower compared to the interior, their orientation distribution did not significantly differ. This can be attributed to the corrective effect of the heat treatment process on fiber orientation at the boundaries, ultimately enhancing the distribution of carbon fibers at the edges and improving the mechanical properties of the composite material. Conversely, the printed model with a raster angle of 45° showed reduced consistency in carbon fiber distribution at the edges. In fiber-reinforced composites produced through FDM printing, fiber alignment at the edges generally surpasses that in the interior [37]. However, heat treatment caused wetting between adjacent fiber materials, resulting in changes in fiber orientation distribution. Furthermore, the temperature of the heat treatment also influenced fiber orientation distribution. When the heat treatment temperature was set at 90 °C, below the glass transition temperature (Tg) of the composite material, black voids significantly increased at the edges of the single fibers, and the consistency of carbon fiber orientation distribution at the fiber edges was relatively poor. Overall, the analysis of the images highlights how heat treatment affects the distribution and orientation of carbon fibers at the edges of the printed models, indicating potential improvements in the composite material’s mechanical properties.

#### 3.2.2. Stereoscopic Fiber Orientation Distribution and Quantitative Analysis

In 3D-printed fiber-reinforced composites, the distribution of fibers on the surface is generally not directly comparable to the orientation distribution of fibers inside. This discrepancy arises due to various factors influencing fiber orientation and distribution during the layer-by-layer printing process. For instance, fibers intertwine between layers, forming a continuous network in both the transverse and longitudinal directions. Additionally, heat treatment can induce changes in fiber orientation, leading to inconsistent distributions between the internal and surface fibers. To accurately assess the fiber distribution in 3D-printed PA6-CF composites, we performed a meticulous process involving the careful removal of the upper and lower surfaces of the obtained monofilament samples. Subsequently, milling and polishing techniques were employed on the exposed specimens to determine the orientation distribution of the internal structural fibers. The polishing process resulted in a remarkably clear orientation distribution of the internal structural fibers, eliminating the need for further treatment. To investigate the effects of heat treatment, the monofilament models printed from the PA6-CF25 composite were subjected to different durations of heat treatment at 120 °C: 2.5 h, 5 h, and 7.5 h. The outcomes of this experiment are presented in Figure 8a–c. It is evident from the results that the number of pores among the monofilaments gradually decreases with increasing heat treatment time, with Figure 8c showing almost no visible pores. This suggests that the process of heat treatment effectively reduces the presence of internal pores in 3D-printed materials, consequently enhancing the mechanical properties of the composite. Similarly, the three-dimensional fiber orientation angles $\theta_{i}$ can be measured, similar to the measurement of planar fiber orientation angles. Using ImageJ 1.53t, we quantified and recorded the angles at which the fibers deviated from the printing direction, along with their respective quantities. This analysis encompassed a total of 1452 visible carbon fibers. The resulting distribution of fiber orientations within the three-dimensional structure is depicted in Figure 9, utilizing a statistical histogram format. The fiber orientation coefficient is expressed using Equation (2) [27].
(2)$$\eta_{0}=\frac{\sum_{i=1}^{n}\cos\theta_{i}}{n}$$
where $\theta_{i}$ is the fiber orientation angle and *n* is the number of visible carbon fibers. In the analysis of fiber orientation distribution, we have also introduced the concept of the fiber orientation distribution fraction $\psi \left(x\right)$, which can be expressed using Equation (3). By calculating this fraction, we are able to quantitatively measure the extent of fiber orientation in the composites. This enables us to conduct a more comprehensive and detailed study and analysis of the properties and structures of fiber-reinforced composites [36].
(3)$$\psi \left(x\right)=\frac{1}{\sigma \sqrt{2\pi }}\int_{-\infty }^{+\infty }e^{-\frac{{\left(x-\mu \right)}^{2}}{2\sigma^{2}}}$$
where μ represents the average orientation angle, and $\sigma^{2}$ denotes the orientation angle variance, these two parameters can quantitatively describe the distribution of fibers. The three-dimensional fiber orientation distribution function obtained through Gaussian fitting is illustrated in Figure 8. The orientation distribution of fibers varies under different heat treatment conditions, resulting in changes in the fitted distribution function. Experimental results demonstrate that the average orientation angle of the 3D-printed PA6-CF25 composite is 0.51 under treatment condition a. This suggests that, under this specific heat treatment condition, the peak frequency of fiber orientation closely aligns with 0° (consistent with the printing direction). Comparatively, the average orientation angle α obtained from fitting under other treatment conditions does not exhibit significant differences, indicating the statistical significance of this one-way analysis method. As the duration of heat treatment increases, the average orientation angle decreases from 0.51 to 0.47 and 0.43, revealing a positive correlation between heat treatment time and the degree of fiber alignment. Longer heat treatment durations result in more uniform fiber alignment, thereby enhancing the tensile strength of the printed composites along the printing direction (i.e., the direction of fiber alignment). Additionally, the variance of the orientation angle signifies differences in fiber orientation distribution among the three heat treatment conditions. Smaller variance values indicate a higher degree of fiber alignment, with the fitted normal distribution curve more concentrated around a fiber orientation angle of 0°. Notably, the sample subjected to a heat treatment temperature of 120 °C for a duration of 7.5 h exhibited the highest degree of fiber alignment, with an α2 value of 8.02.

In summary, the results demonstrate that increasing the heat treatment temperature and duration has a positive effect on fiber alignment, leading to improved mechanical properties of the 3D-printed composite PA6-CF. In the subsequent sections, this study will further investigate the impact of the heat treatment process on the thermal, mechanical, and micro-morphological properties of the printed composites.

### 3.3. Effect of Heat Treatment Process on Other Properties of 3D-Printed PA6-CF Composites

Based on standardized parts with uniform dimensions of 100 mm × 30 mm × 5 mm, we utilized PA6-CF25 composites for 3D printing and subjected them to various heat treatment processes. The density of the composites was measured before and after treatment using Archimedes’ law. Notably, during shearing and extrusion, short-cut carbon fibers tend to create voids, which consequently impact the mechanical properties of the printed samples. To quantify the porosity, we employed Equations (4) and (5), and the porosity was calculated as follows:(4)$$P=\frac{V_{a}-V_{t}}{V_{a}}$$
(5)$$V_{t}=\frac{M\times m_{f}}{\rho_{CF}}+\frac{M\times m_{M}}{\rho_{PA}}$$

Theoretical volume ($V_{t}$), actual volume ($V_{a}$), and actual weight (M) are used to represent the printed specimen’s dimensions and mass. The densities of the carbon fibers ($\rho_{CF}$) and the PA matrix ($\rho_{PA}$) are intrinsic properties of the materials investigated in this study. The composite densities with various carbon fiber contents are displayed in Appendix A. The mass fractions of carbon fibers ($m_{f}$) and the matrix ($m_{M}$) are also considered.

Figure 10b,c illustrates the impact of heat treatment temperature and time on the dimensional change rate and porosity of standard-sized prints. At the heat treatment temperature of 60 °C, the porosity in the printed standard parts showed minimal change (<5%) compared to the untreated samples. As the treatment temperature increased, the porosity in the printed standards decreased significantly. For instance, at 120 °C, the porosity reduced to 6.0%, marking a 15.5% decrease compared to the untreated sample’s 7.1%. However, as the temperature continued to increase to 150 °C, the porosity in the samples began to increase. This phenomenon could be attributed to polymer aging and the introduction of additional defects resulting from excessive temperatures. Additionally, it is worth noting that 120 °C is close to the glass transition temperature of the composites (Figure 10d), at which the movement of polymer chain segments reduces the number of defects. Moreover, increasing the heat treatment time further contributed to reducing the porosity of the printed standards. After 10 h of treatment at 120 °C, the porosity decreased to 5.9%. It is important to mention that the rate of dimensional change in the printed standards, both horizontally and vertically, aligned with the corresponding changes in porosity. The rate of dimensional change in the printed standards decreased from 0% in the untreated samples to 0.6% as the heat treatment temperature and duration increased. Similarly, the porosity of the printed standards also decreased from 7% in the untreated samples to 6%. However, the magnitude of change was more pronounced in the vertical direction, which can be attributed to the increased interlayer porosity resulting from layer-by-layer printing, along with the significant thermal expansion effect observed in the vertical direction during heat treatment. Finally, based on the data presented in Figure 10d, the enthalpy of crystallization of the printed composites increased to 35.23 J/g after undergoing a heat treatment at 120 °C for 10 h, compared to the untreated samples’ 32.24 J/g. The degree of crystallinity also increased from 17.9% to 19.6% when compared to pure PA6 samples. This improvement is attributed to the increased thermal motion energy of the polymer chains due to the higher temperature, resulting in reduced obstruction of chain segment movement and the formation of cross-linked networks. However, excessively high temperatures can also lead to the formation of oxidized networks between the PA6 polymer chain segments, thereby reducing the crystallinity of the composite.

Figure 11 displays SEM micrographs of 3D-printed PA6-CF composites under various heat treatment conditions: untreated, and heat treated at 120 °C for 7.5 h and 120 °C for 10 h. The untreated 3D prints exhibit a relatively large number of voids, whereas the b–c micrographs depict a significant reduction in void quantity. This reduction is attributed to the heat treatment process, which promotes the movement of PA6 chain segments. Macroscopically, this movement is manifested as the penetration and diffusion between neighboring filaments, resulting in reduced void formation between layers and neighboring filaments. The reduction in voids leads to a more uniform stress distribution when the composite is loaded, thereby decreasing the degree of stress concentration and ultimately improving the mechanical properties of the printed composite.

### 3.4. Exploration of Optimal Mechanical Properties of 3D-Printed PA6-CF Composites

After a detailed discussion of the effects of different heat treatment process conditions on the parameters of fiber orientation distribution, thermal properties, and microstructure of the composites, we conducted a systematic investigation. This investigation focused on determining the highest mechanical strength achieved by the 3D-printed PA6-CF composites under the heat treatment process, in both the interfacial and horizontal directions. Based on the results presented in Figure 12, it was observed that under the optimal treatment conditions (i.e., 120 °C for 7.5 h), the printed composites exhibited a tensile strength of 162 MPa and a flexural strength of 175 MPa. In comparison to the untreated samples, the tensile strength showed a remarkable increase of 69%, while the flexural strength increased by 58%. Notably, this tensile strength value for the printed composites is 406% higher than that of the pure PA6 print material. In order to explain how we achieved such high tensile strength values, we used the classical fiber composite material’s tensile strength analysis model to validate our findings. In this model, the tensile strength $\sigma_{c}$ can be expressed using the following formula:(6)$$\sigma_{c}=\eta_{0}V_{f}\sigma_{f}+\sigma_{M}(1-V_{f})$$
where $\eta_{0}$ is the same as in Equation (2), representing the fiber orientation angle, $V_{f}$ is the fiber volume fraction, and $\sigma_{M}$ represents the ultimate stress of the PA6 matrix. The fiber volume fraction can be calculated using the following formula:(7)$$V_{f}=\frac{(1-P)\frac{m_{f}}{\rho_{f}}}{\frac{1}{\rho_{c}}}$$
where $\rho_{f}$ and $\rho_{c}$ are the densities of the fiber and the composite material, and *P* is the porosity. The calculated fiber volume fraction for our PA6-CF25 composite material is 18.4%.

By applying the tensile strength analysis model, we obtained a value of approximately 167 MPa for $\sigma_{c}$ and a value of 0.985 for $\eta_{0}$, which was calculated using our previous CF-OD model. The simulated results closely matched the actual test results, thus explaining why our experimental results significantly exceeded the values reported in the literature. This analysis also demonstrates the significant impact of fiber orientation distribution and fiber content on the mechanical properties of the composite material. 

Furthermore, under the same treatment condition, the interlaminar shear strength (ILSS) of the printed composites reached 42 MPa, indicating a 16.7% improvement when compared to the untreated samples. However, when the temperature exceeds 120 °C, further increasing the temperature causes a decrease in the interlayer adhesion strength of the composite material. This may be due to oxidation cross-linking between molecular chains at high temperatures, resulting in a decline in interlayer mechanical properties. Similarly, increasing the heat treatment duration also enhances the interlayer adhesion strength of the composite material. Figure 13 presents the mechanical performance data of 3D-printed fiber-reinforced polymer composites in recent years. It is evident that the mechanical properties of the printed composites achieved in this study, through process optimization and fine-tuning of printing parameters, greatly surpass the reported average level. The most direct example is from [12], which also uses PA6-CF25 composite material for FDM printing. The reported tensile strength in that study is the highest among all previously reported literature, reaching 105.8 MPa. In this study, further improvement in the strength of the composite material was achieved through process optimization. As a result, we intend to conduct broader research on the potential applications of this material.

### 3.5. Experiments on Lightweighting of 3D-Printed PA6-CF Composites

In our previous section, we provided a detailed account of the progressive enhancement in the mechanical properties of 3D-printed PA6-CF composites. This involved adjusting the carbon fiber content, optimizing printing parameters, and refining heat treatment conditions. Under optimal printing conditions, the composites exhibited highly consistent fiber orientation, a significant reduction in interlaminar and internal voids, and improved material crystallinity. These findings directly correspond to the improved mechanical performance of 3D-printed composites. Building on this existing research foundation, we will now discuss the lightweighting of PA6-CF composites for practical production and real-life applications.

The fiber-reinforced composite material PA6-CF that we have obtained possesses numerous advantages, including high specific strength, good toughness, and low density. Combined with the advantages of additive manufacturing, these materials can be used as support materials for manufacturing automobile crash boxes [43]. In this study, we have referenced the existing literature and designed four honeycomb structures for use as fillers in automobile crash boxes. As shown in Figure 14, we designed Triangular, Hexagonal, Kagome, and Re-entrant as the minimum lattice unit shapes [44]. Using the parameterization method in SolidWorks, we ensured that these four different honeycomb structures had the same density (ρ= 0.126 g/cm^3^). We printed them using two different composite materials, PA6-CF20 and PA6-CF25, and processed them according to the optimal heat treatment conditions to obtain samples. We used durability parameters to evaluate the printed honeycomb structures, including mass (m), initial peak force ($F_{max}$), energy absorption (EA), and specific energy absorption (SEA). Here, Fmax is defined as the maximum force reached during the initial plastic deformation stage, and total energy absorption is calculated by the area under the force-displacement curve, as shown in Equations (6) and (7) [45,46].
(8)$$EA=\int_{0}^{d}P\left(\delta \right)d\delta$$
(9)$$SEA=\frac{EA}{m}$$
where *d* is the total squeezing distance, *P* is the squeezing force, and *δ* is the instantaneous squeezing displacement. *SEA* as an indicator of energy absorbed per unit of mass.

The test results, as shown in Figure 15, demonstrate that the Kagome honeycomb structure exhibits the highest specific energy absorption capability among structures with the same relative density. At a nominal strain of 0.9, the specific energy absorption (*SEA*) of the Kagome structure reaches 17,800 J/kg. This value is comparable to that of metals, despite the PA6-CF composite material having only one-fourth the density of metals. Lightweight, fiber-reinforced nylon materials can effectively absorb and disperse collision energy while reducing the weight of automobile crash boxes. Additionally, compared to metals, high-performance polymer composites are less susceptible to corrosion and oxidation, enabling them to maintain a longer service life in harsh environments. Moreover, fiber-reinforced nylon has lower manufacturing costs. Therefore, manufacturing PA6-CF composite materials under optimized conditions for use as fillers in automobile crash boxes is feasible.

## 4. Conclusions

The inherent layer stacking nature of FDM leads to inconsistent fiber orientation distribution and void formation during the 3D printing process, which hinders further optimization of the composite material’s performance. This study aimed to enhance the performance of 3D-printed PA6-CF composite materials by altering the carbon fiber content, optimizing printing parameters, and refining heat treatment conditions. Through the design of a CF-OD model and microscopic observation of printed composite materials, it was confirmed that heat treatment processes ensure highly consistent fiber orientation within the composite. SEM characterization further confirmed the reduction of interlayer and internal voids, while DSC analysis validated improved crystallinity of the composite material. Comprehensive evaluations of thermal properties, mechanical performance, and microstructural characteristics indicated that under optimal processing conditions, the highest tensile strength of PA6-CF composite material reached 162 MPa, representing a 406% increase compared to pure PA6 material. This represents the greatest reported improvement in the field of CFRP fused-deposition 3D printing to date. Additionally, the study has explored the lightweight applications of the composite material. By implementing 3D-printed PA6-CF composites as fillers in automobile crash boxes, the Kagome honeycomb structure exhibited a specific energy absorption (*SEA*) value of 17,800 J/kg. In comparison to metallic fillers, fiber-reinforced nylon offers advantages such as lightweight design, corrosion resistance, and cost-effectiveness, making it a promising alternative material.

## Figures and Tables

**Figure 1 polymers-15-03722-f001:**
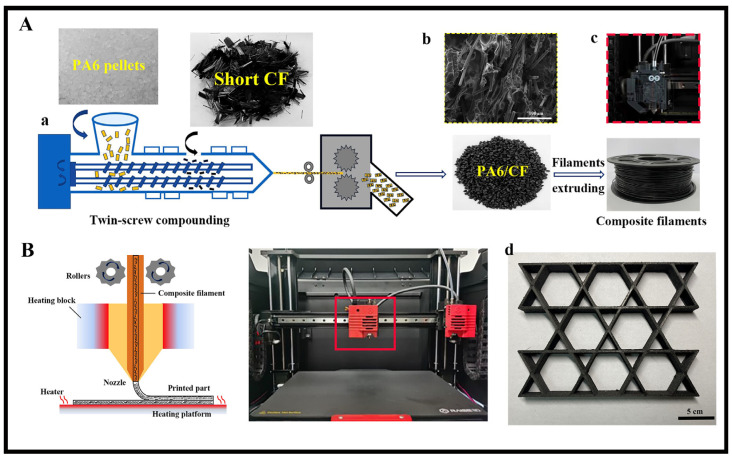
(**A**) Composite PA6-CF filament preparation process; (**a**) Schematic diagram of composite material fabrication; (**B**) schematic diagram of a dual-nozzle composite printer for the preparation of PA6-CF specimens; (**b**) SEM micrographs of PA6-CF particles; (**c**) enlarged view of the internal structure of the printer head; (**d**) printed samples of PA6-CF composite material.

**Figure 2 polymers-15-03722-f002:**
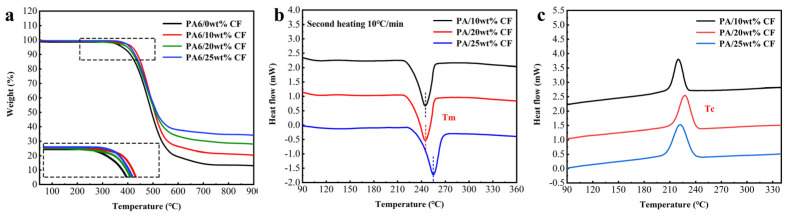
(**a**) TGA curves of PA6-CF composite, the boxed section has been magnified for closer examination; (**b**,**c**) DSC curves of PA6-CF composite.

**Figure 3 polymers-15-03722-f003:**
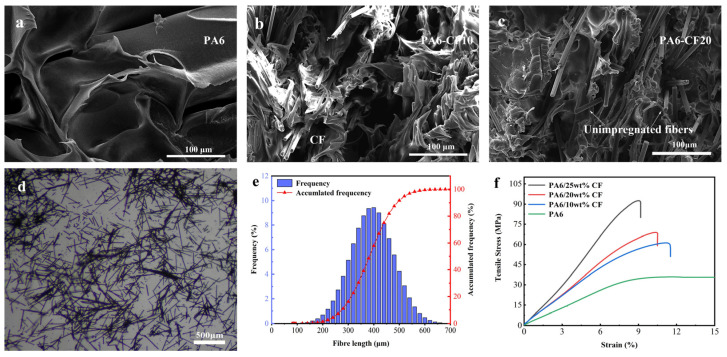
SEM microscopy of composite materials (**a**) PA6; (**b**) PA6-CF10; (**c**) PA6-CF20; (**d**) optical microscope image of residual CF after complete burnout of the composite material; (**e**) fiber length distribution frequency histogram and fiber length distribution cumulative frequency line plot; (**f**) tensile stress–strain curves of composite prints with different short-cut CF contents.

**Figure 4 polymers-15-03722-f004:**
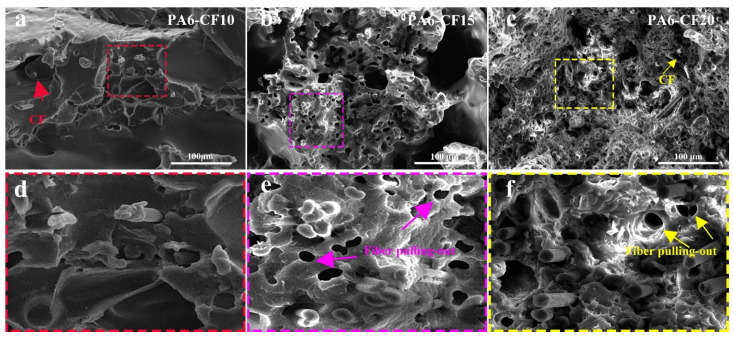
SEM images of tensile fracture surfaces of 3D-printed composites with different CF contents; (**a**) PA6-CF10; (**b**) PA6-CF15; (**c**) PA6-CF20; (**d**–**f**) magnified image of the rectangular area in each test sample.

**Figure 5 polymers-15-03722-f005:**
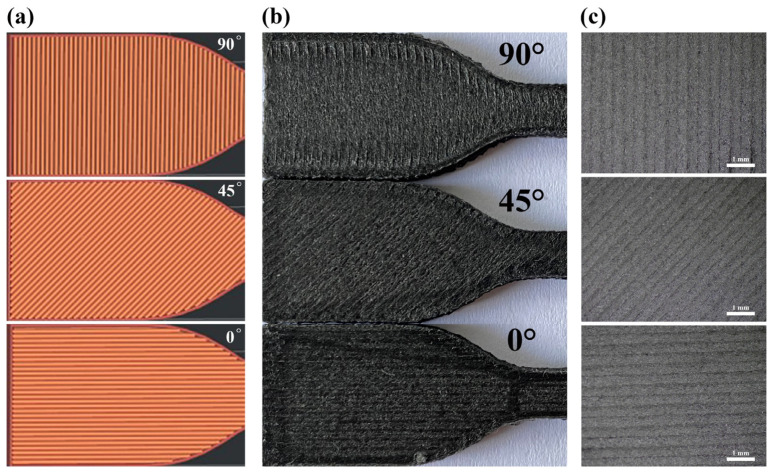
(**a**) Three deposition modes of PA6-CF; (**b**) 3D-printed tensile specimens with different raster angles; (**c**) images of printed samples at different grating angles under an optical microscope.

**Figure 6 polymers-15-03722-f006:**
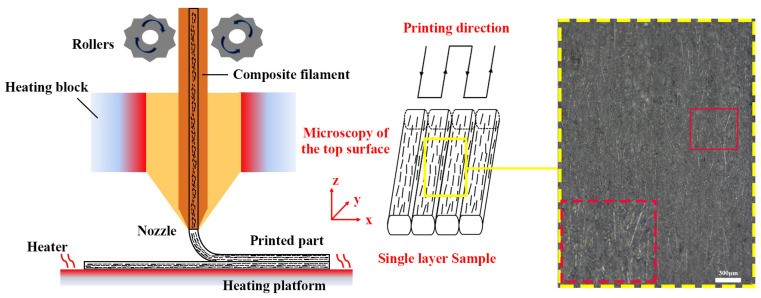
Schematic diagram: preparation and fiber distribution of single-layer monofilament carbon fiber 3D-printed composite sample under microscope, the red dashed box is an enlarged image of the solid line box.

**Figure 7 polymers-15-03722-f007:**
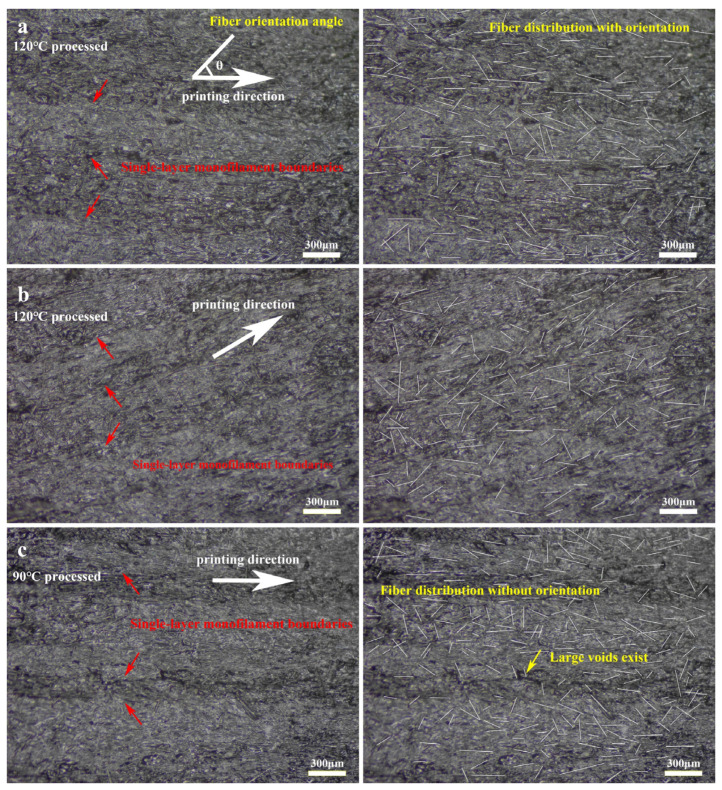
Optical microscopy images of 3D-printed monofilament CF-OD models: (**a**) raster angle 0° printed sample heat treated at 120 °C for 5 h; (**b**) raster angle 45° printed sample heat treated at 120 °C for 5 h; (**c**) raster angle 0° printed sample heat treated at 90 °C for 5 h; carbon fibers on the surface are marked with short white lines.

**Figure 8 polymers-15-03722-f008:**
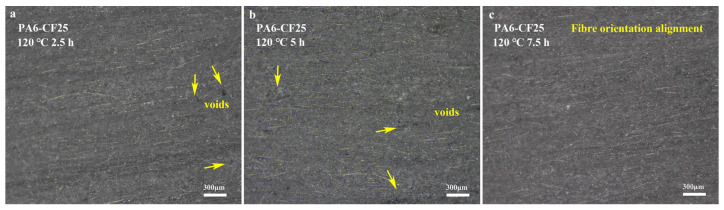
Micrographs of fiber orientation distribution in the internal structure of 3D-printed PA6-CF composites; (**a**) heat treatment at 120 °C for 2.5 h; (**b**) heat treatment at 120 °C for 5 h; (**c**) heat treatment at 120 °C for 7.5 h.

**Figure 9 polymers-15-03722-f009:**
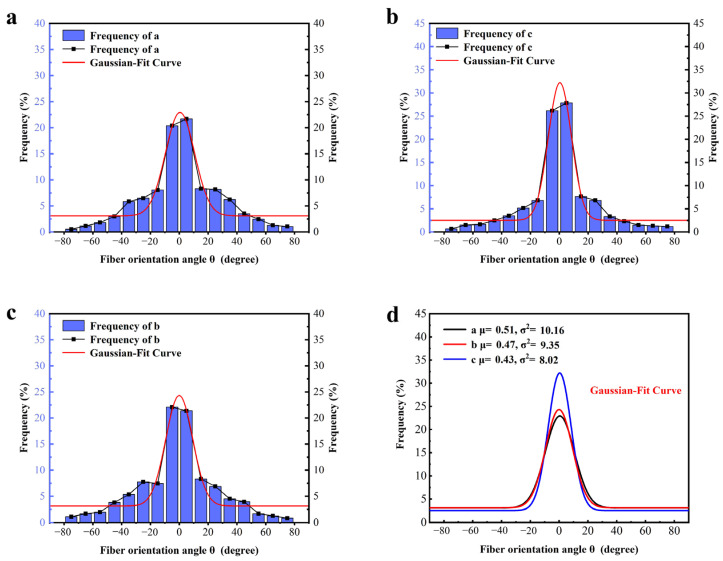
Statistical histogram of fiber orientation distribution and Gaussian-fit normal distribution curve; (**a**–**c**) represent the fiber orientation distribution and distribution curves for three different modes; (**d**) summarizes the fitted curves of fiber orientation distribution under three different processing techniques.

**Figure 10 polymers-15-03722-f010:**
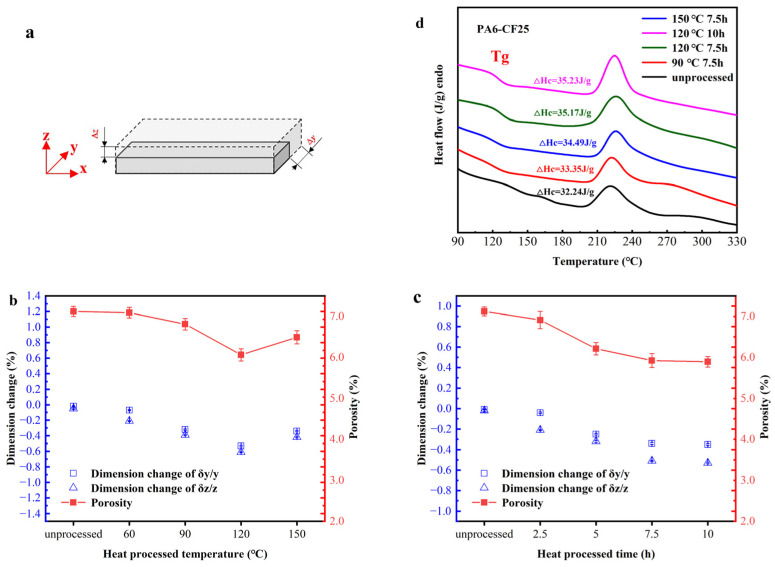
(**a**) PA6-CF composite material 3D printing 100 mm × 30 mm × 5 mm standard specimen; (**b**,**c**) dimensional change rate and porosity of standardized prints under different heat treatment temperature and time conditions; (**d**) DSC curves of the effect of heat treatment on the thermal performance of printed composites.

**Figure 11 polymers-15-03722-f011:**
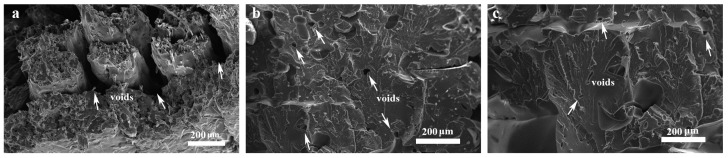
SEM micrographs of void distribution in 3D-printed composites under heat treatment process; (**a**) untreated; (**b**) heat treated at 120 °C for 7.5 h; (**c**) heat treated at 120 °C for 10 h.

**Figure 12 polymers-15-03722-f012:**
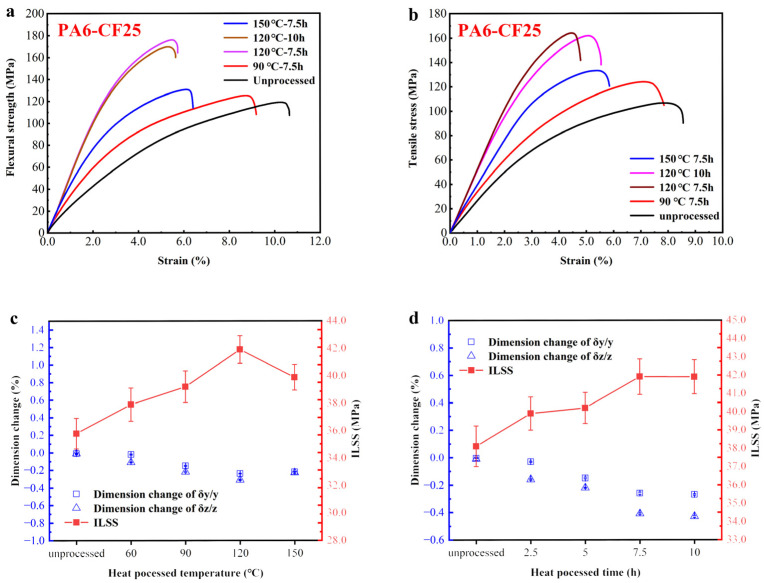
(**a**) Stress–strain curve of flexural test after heat treatment of 3D-printed PA6-CF25; (**b**) Stress–strain curve of 3D-printed PA6-CF25 tensile test after heat treatment; (**c**,**d**) ILSS and dimensional changes of printed PA6-CF25 under different heat treatment conditions.

**Figure 13 polymers-15-03722-f013:**
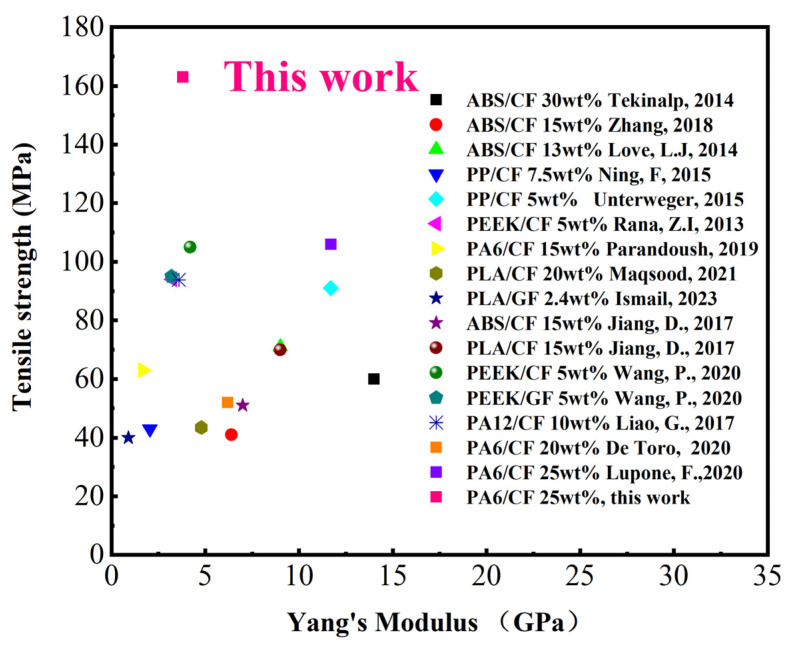
Summary of mechanical performance data for 3D-printed fiber-reinforced polymer composites. The references in order of citation are as follows: [4,5,6,9,12,13,15,16,19,24,29,32,34,36].

**Figure 14 polymers-15-03722-f014:**
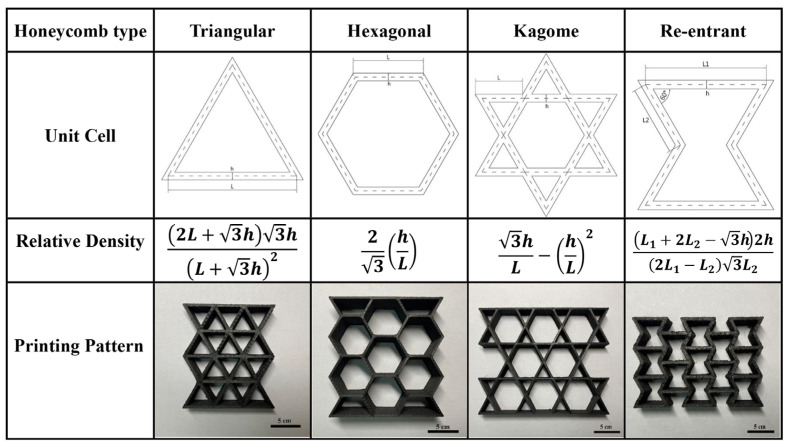
Unit cell structure, relative density, and printing results of four different honeycomb structures.

**Figure 15 polymers-15-03722-f015:**
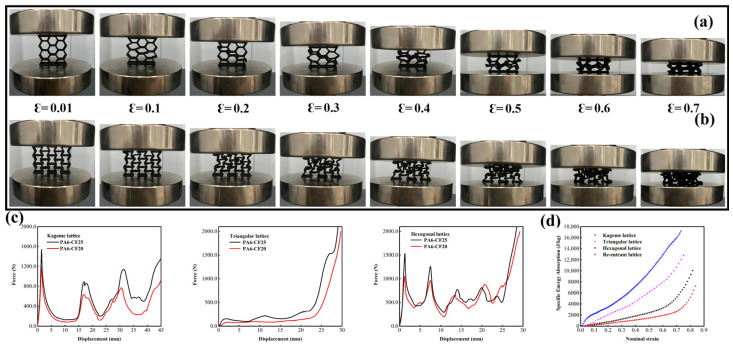
Examples of representative lattices; (**a**) compression experimental plots of printed samples of PA6-CF25 Hexagonal lattice; (**b**) compression experimental plots of printed samples of PA6-CF25 re-entrant lattice; (**c**) compression force-displacement curves of several typical honeycomb lattices; (**d**) images of *SEA* values of four lattice structures as a function of nominal strain.

**Table 1 polymers-15-03722-t001:** Printing parameters of the E2CF printer and heat treatment parameters.

Parameters	Value	Parameters	Value
Materials	PA6, PA6-CF10, PA6-CF20, PA6-CF25	Build plate temperature	60 °C, 90 °C
Infill structure	Triangular, Hexagonal, Kagome, Re-entrant	Layer thickness	0.15 mm
Printing speed	20 mm/s	infill density	100%
*X/Y* axis movement speed	150 mm/s	Nozzle temperature	275 °C
Extrusion line width	0.4 mm	Heat treatment temperature	rt, 90 °C, 120 °C, 150 °C
Raster angle	0°, 45°, 60°, 90°, 135°	Heat treatment time	0 h, 5 h, 7.5 h, 10 h

**Table 2 polymers-15-03722-t002:** Mechanical test results of printed samples with varying parameters.

Sample (Fiber Contents/Raster Angle/Build-Up Temperature)	Tensile Strength (MPa)	Young’s Modulus (MPa)	Flexural Strength (MPa)	Impact Strength (kJ/m^2^)	Elongation at Break (%)
10 wt%CF angle 135° T90 °C	55.9 ± 1.9	489.3 ± 4.5	108.2 ± 1.3	11.8 ± 1.1	13.4 ± 0.2
10 wt%CF angle 90° T90 °C	46.8 ± 2.8	424.5 ± 7.6	115.2 ± 1.9	9.2 ± 0.4	13.2 ± 0.2
10 wt%CF angle 45° T90 °C	57.3 ± 1.8	492.3 ± 4.1	109.3 ± 0.9	11.1 ± 0.3	13.2 ± 0.1
10 wt%CF angle 0° T90 °C	59.1 ± 2.0	513.5 ± 3.1	117.2 ± 2.1	13.1 ± 0.1	14.3 ± 0.5
20 wt%CF angle 135° T90 °C	60.3 ± 2.3	582.8 ± 5.6	124.2 ± 1.3	15.1 ± 1.4	10.6 ± 0.2
20 wt%CF angle 90° T90 °C	57.3 ± 2.5	551.2 ± 4.5	129.1 ± 1.7	12.1 ± 2.5	10.1 ± 0.1
20 wt%CF angle 45° T90 °C	61.2 ± 1.8	560.2 ± 3.9	121.2 ± 2.3	15.2 ± 0.3	11.2 ± 0.4
20 wt%CF angle 0° T90 °C	65.6 ± 2.0	604.2 ± 3.9	110.2 ± 1.1	14.1 ± 0.8	11.8 ± 0.5
25 wt%CF angle 135° T60 °C	64.1 ± 2.5	614.0 ± 5.1	114.1 ± 1.2	17.2 ± 0.2	11.2 ± 0.3
25 wt%CF angle 90° T60 °C	61.1 ± 1.2	568.9 ± 4.9	132.2 ± 2.1	15.1 ± 0.7	11.6 ± 0.4
25 wt%CF angle 45° T60 °C	65.2 ± 1.9	616.1 ± 4.5	113.5 ± 0.7	15.4 ± 0.5	11.2 ± 0.7
25 wt%CF angle 0° T60 °C	71.0 ± 2.3	641.2 ± 5.9	111.2 ± 1.5	17.3 ± 0.4	12.0 ± 0.2
25 wt%CF angle 135° T90 °C	81.3 ± 1.7	791.01 ± 4.5	124.3 ± 2.4	24.0 ± 0.8	9.3 ± 1.0
25 wt%CF angle 90° T90 °C	77.7 ± 2.1	711.48 ± 4.1	145.2 ± 2.3	26.1 ± 0.6	10.55 ± 0.9
25 wt%CF angle 60° T90 °C	86.2 ± 1.1	877.2 ± 3.7	126.4 ± 3.1	23.1 ± 1.1	9.8 ± 1.1
25 wt%CF angle 45° T90 °C	85.3 ± 1.0	862.38 ± 3.3	135.1 ± 1.3	21.1 ± 0.3	9.3 ± 0.4
25 wt%CF angle 0° T90 °C	97.7 ± 2.1	1021.8 ± 9.1	123.1 ± 0.7	25.1 ± 0.2	11.7 ± 1.3

## Data Availability

Not applicable.

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
