# Peer review of "Fused-Deposition Modeling 3D Printing of Short-Cut Carbon-Fiber-Reinforced PA6 Composites for Strengthening, Toughening, and Light Weighting"

_polymers, 2023, doi:10.3390/polym15183722_

Round 1

Reviewer 1 Report

This paper focuses on the FDM 3D printing of PA6 reinforced with short-cut carbon fibers. The effect of carbon fiber content, heat treatment and printing parameters on the micro-morphology, thermal and mechanical properties of the composites were studied. Finally, the authors developed an application for the proposed composite in automobile crash boxes. The article title is very interesting and new, the images are wonderful and the article is well organized.

Revision comments:

1.     The abstract can be made more attractive by using more quantitative data. It is suggested to provide more quantitative results.

2.     The novelty and purpose of the research should be clearly stated in the abstract and introduction.

3.     There are some short forms without further details, like in the abstract section that mentions CFRP, but the authors did not explain what it stands for.

4.     Some parts of the article, such as the introduction, should be revised for consistency. So, the introduction should be written better and needs minor revisions. The part of the first and second paragraphs of the introduction is general information and can be stated more briefly.

5.     Use the following resources to deepen the introduction. Shape memory performance of PETG 4D printed parts under compression in cold, warm, and hot programming. Development of Pure Poly Vinyl Chloride (PVC) with Excellent 3D Printability and Macroand MicroStructural Properties. 3D Printed High Performance 4D printing of PET-G via FDM, including tailormade excess third shape.

6.     Figure 2 was not mentioned in the manuscript.

7.     Figure 3 (f) is unrelated to the other pictures presented in Figure 3.

8.     How are the printing parameters selected? How is the printing quality confirmed and checked? How is the reproducibility of experimental test results checked?

9.     Figure 5 (a) indicates one layer for wall thickness, but in Figure 5 (b), printed specimens have at least two layers of wall, which can influence the mechanical properties.

10. There are some mismatches between figure numbers and their description in the manuscript. For instance, Figure 6 (d) is mentioned on page 7 but does not match Figure 6.

11. The quality of Figure 15 is poor, making it hard to read.

***

Reviewer 2 Report

The article "Fused deposition modeling 3D printing of short-cut carbon fiber  reinforced PA6 composites for strengthening, toughening, and light weighting" presented the roles of carbon fiber (CF) in FDM printed CF-Nylon composites. The works are fall within the scope of Polymers. The amount of works done is impressive but the presentation/results & discussion can be further exhanced. Various factors were considered in this study and presented in current manuscripts. 

To improve the readability and quality of the manuscript, the authors are advised to

1) focus on fewer objectives OR present the results (for example results in Table 2) in a more systematic way. Currenlty too many information was presented in Table 2 and readers are unable to extract useful information from table 2. Authors can consider to use DOE or ANOVA.

 2) Figure 1. B- Fused deposition process, i believe the figure 1B is not a fused deposition process. 

3) 2. Materials and Methods, more details for heat treatment, measurement of dimensional accuracy should be provided.

4) Table 1: honeycomb structure- should it be 'infill structure'?

5) Figure 5b, please label the raster angle, or provide a clear image. Cannot really see the raster angle from your figure.

6) lines 390-391- Longer heat treatment durations result in higher fiber alignment, thereby enhancing the mechanical properties of the printed composites -

- >How about the adhesion between layers? Did you check with samples without CF? The improve in mechanical properties might caused by the improved in interlayer adhension when the thermoplastic subjected to heat treatment.

7) Lines 441-442- it will be great if the results of dimensional accuracy can be presented. 

8) Fig 13, please check 'Yang's modulus'. 

Also, please update the fig with some recent works (2021-2023), here are some examples but not limited to 

https://doi.org/10.1021/acsami.2c02076
https://doi.org/10.3390/polym15163436
https://doi.org/10.1016/j.jcomc.2021.100112

DOI 10.1088/1757-899X/322/2/022012

Please also discuss the results from ref 5 and compare with your results. Both are PA6/25 wt%. 

9) General
The title aims to focus on "
strengthening, toughening, and light weighting",
but too many sub- topics were presented in this 20-page article. Please consider to focus on the key messages.

Reviewer 3 Report

Dear Authors, after reading your manuscript, I think that your manuscript could be reconsidered for publication should you be prepared to incorporate major revisions.

Abstract-the originality and novelty of the research activity are not well exposed and highlighted. What is the motivation for this study?

The introduction was too long and had lots of irrelevant information. The first two paragraphs in Introduction is very superficial and general information. The importance of 3D printing and additive printing methods is not hidden to anyone. The main contribution of this work was the composite composed of PA6/CF. The introduction should focus on the FDM printing PA6-CF which is the subject of the manuscript. What were the latest achievements of PA6 mixing CF? What were the limitations in these published papers? What were the unique contributions of this job? These details needed to be offered and discussed in Introduction.

Materials and Methods- please describe your selected methods. Why are these best as the others? Why did you choose PA6 material? Testing standard need to be provided.

Please make sure your conclusions' section underscores the scientific value-added of your paper, and/or the applicability of your findings/results. Highlight the novelty of your study.

In addition to summarizing the actions taken and results, please strengthen the explanation of their significance. It is recommended to use quantitative reasoning comparing with appropriate benchmarks, especially those stemming from previous.

Best regards

Round 2

Reviewer 2 Report

The authors have revised the article based on previous comments. However, minor spelling errors and grammatical errors are still detected.

For example, Fig 13 Yang's modulus

minor spelling errors and grammatical errors are still detected.

For example, Fig 13 Yang's modulus

Reviewer 3 Report

Dear Authors, thanks for considering and discussing all my mentioned points. I see a significant improvement in your manuscript and I recommended to accept the paper.

Thanks for sharing your research.

Best regards.